# Recent Advancements in the Development of Nanocarriers for Mucosal Drug Delivery Systems to Control Oral Absorption

**DOI:** 10.3390/pharmaceutics15122708

**Published:** 2023-11-30

**Authors:** Hideyuki Sato, Kohei Yamada, Masateru Miyake, Satomi Onoue

**Affiliations:** 1Laboratory of Biopharmacy, School of Pharmaceutical Sciences, University of Shizuoka, 52-1 Yada, Suruga-ku, Shizuoka 422-8526, Japan; h.sato@u-shizuoka-ken.ac.jp (H.S.); k.yamada@u-shizuoka-ken.ac.jp (K.Y.); 2Business Integrity and External Affairs, Otsuka Pharmaceutical Co., Ltd., 2-16-4 Konan, Minato-ku, Tokyo 108-8242, Japan; ma-miyake@hanshin-group.co.jp

**Keywords:** mucodiffusion, mucus layer, nanocarriers, oral absorption, surface properties

## Abstract

Oral administration of active pharmaceutical ingredients is desirable because it is easy, safe, painless, and can be performed by patients, resulting in good medication adherence. The mucus layer in the gastrointestinal (GI) tract generally acts as a barrier to protect the epithelial membrane from foreign substances; however, in the absorption process after oral administration, it can also disturb effective drug absorption by trapping it in the biological sieve structured by mucin, a major component of mucus, and eliminating it by mucus turnover. Recently, functional nanocarriers (NCs) have attracted much attention due to their immense potential and effectiveness in the field of oral drug delivery. Among them, NCs with mucopenetrating and mucoadhesive properties are promising dosage options for controlling drug absorption from the GI tracts. Mucopenetrating and mucoadhesive NCs can rapidly deliver encapsulated drugs to the absorption site and/or prolong the residence time of NCs close to the absorption membrane, providing better medications than conventional approaches. The surface characteristics of NCs are important factors that determine their functionality, owing to the formation of various kinds of interactions between the particle surface and mucosal components. Thus, a deeper understanding of surface modifications on the biopharmaceutical characteristics of NCs is necessary to develop the appropriate mucosal drug delivery systems (mDDS) for the treatment of target diseases. This review summarizes the basic information and functions of the mucosal layer, highlights the recent progress in designing functional NCs for mDDS, and discusses their performance in the GI tract.

## 1. Background

Oral delivery is a desirable route of administration for various types of drugs since it is easy to use, noninvasive, painless, economical, and administered by patients [1]. These advantages can improve patient adherence, possibly leading to the achievement of the optimal effectiveness of medications [2]. Despite the various advantages of the oral route for drug administration, several factors, such as external barriers in the gastrointestinal (GI) tract, make it challenging to control and estimate the oral absorption process of target drugs. The major obstacles to oral absorption in the GI tract are the severe pH gradient from the stomach to the colon, metabolic enzymes, the mucus layer on the surface of epithelial cells, and the epithelial cellular membrane [3]. Although these physiological functions are essential for maintaining homeostasis in the human body by degrading and eliminating exogenous materials that have the potential to be harmful, the available amount of administered drug can also be influenced, potentially lowering the oral bioavailability and efficacy and increasing the need for more frequent dosing. Thus, appropriate oral delivery systems that do not destroy the barrier systems should be developed for effective and safe medications.

To overcome these barriers during absorption, different types of drug delivery systems have been developed with a focus on designing nanocarriers (NCs). There have been several reports on the development of NCs including liposomes, solid lipid nanoparticles, nanostructured lipid carriers, polymeric micelles, polymeric nanoparticles, inorganic nanoparticles, and so on [4,5,6,7]. Generally, NCs can be developed to achieve efficient drug delivery by (i) improving dissolution behavior by increasing the active surface area, (ii) stabilizing inner compounds by encapsulation, (iii) controlling the release of encapsulated drugs, (iv) changing the diffusive properties within the mucus layer (addition of mucopenetration and mucoadhesion properties) at the absorption site, and (v) enhancing intestinal cellular uptake [7,8,9,10]. Most of the reported conventional NCs for oral DDS mainly focus on the enhancement of dissolution properties and controlled release of encapsulated drugs; however, those might not be sufficient to achieve pharmacokinetic control of the absorption process owing to various physiological barriers in GI tracts. Although many factors affect the physicochemical properties of NCs, their potential depends mainly on the surface properties that determine their fate in the GI tract, as the surface is always exposed to the harsh environment of the GI tract [5]. Therefore, the development of suitable surface design technologies is a key consideration for the highly efficient oral DDS.

Recently, mucosal drug delivery systems (mDDS) have been investigated for the oral administration of pharmaceutical agents [11]. The mucus layer is one of the ubiquitous systems; a viscous layer covers epithelial cells in many parts of the body [12]. The mucus layer mainly consists of mucin proteins, which are clustered into highly glycosylated and non-glycosylated mucin domains [13]. In the intestinal tract, it acts as a lubricant and traps pathogens and other undesired xenobiotics [14], whereas this protection mechanism can also cause a reduction in the bioavailability of orally dosed drugs. The mucus layer can be used to adjust the residence time of NCs by modifying the surface properties to develop mDDS-based NCs to control intestinal absorption. NCs with mucoadhesive and mucopenetrating potentials can be developed by changing the interactions between the mucin layer and the surface of NCs (Figure 1). These properties could contribute to the control of the absorption process of drugs encapsulated in the NCs after oral administration. Generally, mucoadhesive NCs can extend the absorption process, which results in prolonged systemic exposure, and mucopenetrating NCs can achieve quick absorption from the absorption site. This review briefly summarizes the properties of GI mucus that make it attractive for controlling the oral absorption of drugs and describes the surface properties of NCs that impact the interaction with mucus layers.

## 2. Characteristics of Mucosal Layer in GI Tract

### 2.1. Physiological Functions of Mucus Layer

The main constituents of mucus are water (90–95%), electrolytes, lipids (1–2%), and proteins [13]. Owing to the presence of mucin, a large complex glycosylated protein, mucus can form mesh-like structured viscous gel layers on various mucosal tissues, such as the GI tract, eyes, nose, and respiratory tract [15]. There are two types of mucins: membrane-bound mucins and secreted (gel-forming) mucins, and mucus layers are composed of gel-forming mucins secreted from goblet cells [16]. Mucin 2 (MUC2) is the main component of intestinal mucus and forms the mucus skeleton. The structure of mucin includes sulfate groups on *N*-acetyl glucosamine and galactose and carboxylic groups on sialic acid sugars, providing an overall negative charge to mucins under most pH conditions [17]. The surface of epithelium in the GI tract is covered by mucus, which consists of mucin polymers connected via disulfide bonds, forming mucus layers. Mucins are continuously secreted from goblet cells in the GI tract, and the thickness of the mucus differs depending on the balance between its production and turnover [18]. The mucus layer is thinnest in the intestinal tract and thickest in the stomach and colon. The mucus layer of the small intestinal tract contains a high concentration of peptides and proteins with antibacterial activity that contribute to the removal of bacteria [19]. Because the risk of infection in the small intestine is higher than that in other parts of the GI tract, such as the stomach and colon, these protective functions are very important.

In the GI tract, the mucus layers can act as barriers to protect the surface of epithelial cells from foreign materials with harmful potentials and pathogens by trapping them in the mesh structure and disturbing their diffusion towards the epithelium [12]. There are two possible mechanisms of mucosal barrier systems: (i) size exclusion by mucin mesh-like structures and (ii) molecular interactions between mucin and drugs, including electrostatic and hydrophobic interactions. The mucus barrier is a high-density mucin fiber network with an average pore size of 20–200 nm [20]. Therefore, the mucus layer acts as a biological sieve. Small molecules such as nutrients, water, and gas can pass through the mesh structure, whereas particles larger than the pore size of the mesh structure experience steric hindrance and can be trapped by the structure. Mucus has significant blocking effects on molecules with a molecular weight of 30,000 Da [21]. This size-exclusion mechanism also protects the epithelial membrane from bacteria and foreign particles (>0.5 μm) [22], contributing to the maintenance of a sterile environment around the surface of the epithelium. Theoretically, the smaller the number of drug molecules/particles, the easier it is for them to penetrate the mucosal layer. However, even if the particles are much smaller than the pore size of the mucin mesh, molecular interactions between mucin can impair the diffusion properties of drug molecules/particles by significantly increasing the solute-solvent resistance [23]. Nonpolar solvents, such as oils, diffuse more slowly through the mucus than through water because of the hydrophobic interactions in the lipophilic contents of the mucin layer. The lipid content in the mucus layer can form hydrophobic interactions between mucus and diffusing drug particles, even those smaller than the pore size. In addition, as described above, there are many sulfate and sialic acid moieties in the mucin structure that create a strong negative charge on its surface. Therefore, electrostatic interactions can form between charged particles and the mucus layer. Cationic molecules such as chitosan, a natural polysaccharide with mucoadhesive properties, can form tight polyvalent bonds with negatively charged moieties in mucin [24].

The continuous secretion of mucus not only prevents pathogens and foreign substances from entering the epithelial membrane but also removes various compounds and drug molecules. Thus, appropriate drug delivery systems that are based on clearance mechanisms of mucus systems should be considered to achieve sufficient oral absorption.

### 2.2. Roles of Mucin in Mucopenetrating and Adhesive Formulations

The mucus layer on the surface of the epithelial membrane can act as a smart physiological barrier not only for foreign substances with harmful potential and pathogens but also for orally dosed drugs. For effective and sufficient oral drug delivery, avoiding protective mechanisms and/or even turning barrier mechanisms should be considered. Therefore, several strategies have been developed to control the diffusive properties of drug nanoparticles within the mucus layer, including the mucopenetration and mucoadhesion of NCs.

Mucopenetrating NCs can achieve efficient oral delivery of target drugs with higher amounts of oral absorption and subsequently improve oral bioavailability, as this system can deliver the carrier cargo close to the absorption site in the GI tract [25]. As described in Section 2.1, various interactions, including entanglement with mucin, electrostatic interactions, and hydrophobic interactions can trap foreign substances and prevent NC penetration through the mucus layer. To obtain mucopenetrating properties, minimizing the interactions between NCs and mucin, that is, creating a bioinert surface, is important [5]. Entanglement is the biggest barrier to the penetration of NCs; thus, decreasing entanglement would enable NCs to move more easily through the mucus layer. Hydrogen bonds and ionic interactions can be formed between NCs with a high charge density and negatively charged sialic acid groups in the mucus structure. Thus, reducing the net charge and charge density can suppress these interactions, possibly resulting in a more bioinert surface against the mucus layer. To reduce the net charge on the surface of NCs, previous studies report covering the NC with uncharged materials or highly densely charged materials with evenly distributed positive and negative charges [26].

Mucoadhesive NCs have also attracted considerable interest in controlling and prolonging the residence time of NCs at the absorption sites in the GI tract. Mucoadhesion is a complex phenomenon involving various types of adhesion mechanisms, including physical entanglement, dehydration, electrostatic interactions, covalent bonds between thiol groups in mucin, and multiple low-affinity bonds, such as hydrogen bonds and van der Waals forces [27]. There are two main mechanisms of mucoadhesion: contact and consolidation [28]. In the first step, the material must be in close contact with the mucus layer surface. If the attractive forces (van der Waals forces and electrostatic attraction) between the materials and the mucus layer are not strong enough to overcome the repulsive forces (e.g., osmotic pressure and electrostatic repulsion), the adhered particles can be easily removed by GI motions and physiological turnover of the mucus layer. Consolidation is also necessary to prolong the adherence of the NCs to the mucus layer. This process can strengthen the interactions between NCs and the mucin, possibly leading to resistance to the clearance mechanisms of NCs from the mucus layer. The consolidation process has been explained by two different theories: the interpenetration theory and the dehydration theory. According to the interpenetration theory, the glycoproteins of mucin and mucoadhesive compounds should closely interact by the interpenetration of their chains and the formation of secondary bonds, contributing to an increase in both chemical and mechanical interactions [29]. According to the dehydration theory, when mucoadhesive compounds with gel-forming properties are in contact with the mucus layer, the material can induce dehydration of the mucus due to different osmotic pressures. Until the osmotic pressure is equilibrated between the material and mucus, different concentration gradients cause water movement. The dehydration process enhances the mixing of the material and mucus, resulting in increased contact time with the mucus membrane. Generally, polysaccharides, including chitosan, alginate, and cellulose derivatives, have been reported as mucoadhesive polymers and are used as carrier materials for mucosal drug delivery systems.

Understanding the appropriate interactions and mechanisms of the penetration and/or adhesion of NCs in the mucus layer has enabled researchers to identify, select, and develop materials for designing functional NCs. In the next section, conventional materials with mucoadhesive and mucopenetrating potential and their recent applications for designing NCs are described.

## 3. Controlling the Diffusion Properties of NCs in the Mucus Layer

### 3.1. Mucopenetrating Nanoparticles

Mucopenetrating nanoparticles can diffuse through the mucus layer and quickly reach the epithelial membrane (absorption site) of the GI tract. This characteristic could give NCs potential advantages in the oral delivery of environmentally sensitive drugs such as peptides and proteins because of the penetrating ability of the mucus and the release of the inner drugs at the epithelium rather than within the lumen. The generation of a bioinert surface is a key consideration in the design of mucopenetrating particles. In this section, the main strategies for obtaining NCs with bioinert surfaces are summarized, including polyethylene glycol (PEG), zwitterionic (virus-mimicking), and other strategies, such as mucolytic surfaces (Table 1).

#### 3.1.1. Polyethylene Glycol (PEG)-Coated Surface

PEG is a widely known bioinert and highly biocompatible hydrophilic polymer with a chemical structure of (CH_2_CH_2_O)*_n_*. Thus, PEG surfaces are broadly used to provide bioinert characteristics not only in the GI tract but also in the bloodstream for various DDS approaches. Owing to their highly hydrophilic characteristic, PEG chains can form a dense hydrated brush on the surface of NCs, protecting them from enzymatic degradation and interactions between GI fluids and mucosal components [42]. The neutral charge and high mobility of the PEG chain can also contribute to minimizing electrostatic interactions with mucins, providing easier penetration through the mucus layer [11]. There have been several reports on the development of NCs with PEG surfaces owing to their high biocompatibility and bioinertness [30,31]. The chain length, bush density, and architecture of PEG significantly influence the mucopenetrating characteristics of the NCs [42]. In many reports, a PEG chain length of >2000 Da is frequently used [33,42,43,44]. Mert et al. found that PEG 5k-coated poly(lactic-*co*-glycolic acid) (PLGA) showed significantly higher penetration within the mucus than PEG 1k-coated PLGA nanoparticles [34]. Further, Inchaurraga et al. demonstrated significantly higher mucopenetrating properties of PEG 2 kDa- and 5 kDa-coated NCs, consisting of a copolymer of methyl vinyl ether and maleic anhydride, than those coated with PEG 10 kDa [45]. These reports indicate that there is an optimal range of the length of PEG chains and long PEG chains that may entangle the mucus network, resulting in impaired mucopenetration. With respect to the density of the PEG brush on the surface of the NCs, a high-density PEG coating is beneficial for achieving mucopenetration compared to a loose blush of the PEG chain [46]. The architectures of PEG chains, including bottles [47], branched [48], cross-linked [49], and looped PEG [50], also have highly bioinert characteristics, possibly contributing to the design of bioinert surfaces for NCs.

Poloxamers, which are copolymers containing PEG and poly(propylene glycol), have also been widely investigated for the design of NCs with PEGylated surfaces. Poloxamers contain both hydrophilic and hydrophobic blocks in their structure; thus, they can be incorporated into hydrophobic NCs such as PLGA nanoparticles, self-emulsifying drug delivery systems (SEDDS), solid lipid nanoparticles, nanostructured lipid carriers, and liposomes to improve the dispersibility and bioinertness of NCs [32,51]. Additionally, poloxamers are non-ionic surfactants; therefore, their electrostatic interactions are limited.

According to previous reports, the PEG-coated surface enables NCs to penetrate the mucus layer quickly; however, a highly bioinert surface also reduces the interaction with the cellular membrane at the absorption site, thereby possibly limiting the absorption process if free drug molecules cannot be released from the NCs. Additionally, the cellular uptake of PEG-coated NCs, especially those coated with long PEG chains, can be limited because of steric hindrance, reduced surface charge, and hydrophilic–hydrophobic repulsion [52]. Therefore, it is necessary to consider the balance between the bioinertness with mucin and its interactions with cellular membranes to control the absorption process using PEG-coated NCs. In recent years, there have been some reports on the appearance of PEG antibodies due to the excessive use of PEG for DDS applications [53], thereby causing the alteration of mobility and biodistribution of PEGylated nanoparticles in mucus [54]. Thus, developing an alternative bioinert polymer might be necessary for designing desirable oral NCs.

#### 3.1.2. Zwitterionic (Virus-Mimicking) Surface

Certain viruses, such as the Norwalk virus, Hepatitis B, and human papillomavirus, can diffuse within the mucus layer as quickly as in aqueous or saline solutions [55,56]. Thus, designing the surface of NCs to mimic a virus is a promising strategy for achieving sufficient mucopenetration [20]. The major characteristic of viruses is a high density of charged surfaces with equal amounts of both anionic and cationic components, resulting in neutrally charged bioinert surfaces [57]. The densely charged surface can prevent nonpolar interactions with mucin, owing to a reduction in the exposure of the hydrophobic domains of the virus surface to the mucus layer. As described in Section 2.2, the neutral net charge of the viral surface can make the surface bioinert.

In addition to the densely charged neutral surface, the zwitterionic surface can enhance the hydration of the NC surface due to its highly polar properties, forming ion-dipole interactions and hydrogen bonding with the surrounding water molecules. Such surface properties enable the immobilization of water molecules on the surface of NCs, forming a stable water layer that protects them from interactions with components in the GI tract. The water-binding properties of a zwitterionic sulfobetaine substructure unit can bind 7–8 water molecules, whereas an ethylene glycol unit consisting of PEG chains can bind only one water molecule [58]. This hydration shell layer may contribute to the enhanced mucopenetrating properties of the NCs. Polycarboxybetaine-coated NCs exhibited a 6.7-fold higher mucopenetrating ability than PEG-coated NCs [38]. To prepare NCs with zwitterionic surfaces, amphipathic materials such as phospholipids, polycarboxybetaine, polyphosphorylcholine, polysulfobetaine, and polydopamine can be used as coating materials [35,36,37,39,59].

Apart from amphipathic polymers, the combined use of cationic and anionic materials is another option for designing a neutral NC surface with a highly dense charge. Anionic polymers such as alginate, PAA, hyaluronic acid, chondroitin sulfate, pectin, and carrageenan, and cationic polymers such as chitosan, protamine, and polymethacrylates with amino or ammonium substructures have been reported to develop multi-penetrating NCs in combination. The combined use of chitosan and chondroitin sulfate for the preparation of mucopenetrating NCs exhibited higher mucodiffusive properties than control PLGA nanoparticles, and there have been some reports on the application of other combinations of anionic and cationic polymers [60]. Despite its increasing number of applications on zwitterionic NCs in recent years, the potential for mucopenetration remains unexplored, and possible mechanisms are still unclear. Thus, a deeper understanding of mucopenetrating mechanisms by zwitterionic NCs could help select the appropriate components in functional NCs.

#### 3.1.3. Mucolytic Strategies

Mucopenetrating systems are generally based on making the particle surface bioinert to reduce interactions between the surface of NCs and mucus components. On the other hand, the mucolytic system, also known as the active mucopenetrating system, is another strategy for achieving effective oral delivery through the GI mucus layer [11]. There are two main strategies for designing mucolytic NCs: breaking the disulfide bonds by mucolytic drugs or mucolytic enzymes. Although cleavage of the mucosal layer leads to improved oral absorption of the target drug and drug nanoparticles, there are concerns regarding the risk of pathogen diffusion due to the disruption of the mucus layer. Thus, an appropriate system should be considered to achieve localized degradation of the mucus layer around the absorption site.

N-acetyl cysteine (NAC), a sulfhydryl compound with a free sulfur group, is a mucolytic agent because of its ability to form disulfide bonds with cysteine groups in the mucus layer and is clinically used as an expectorant drug [25]. Thus, NAC can cleave disulfide bonds in the mucus layer and reduce the cross-linking of mucus gels, possibly leading to enhanced mucopenetration of NCs [25,40]. Dithiothreitol, thiobutylamidine-dodecylamide, and thioglycolic acid-octylamine have also been investigated as mucolytic agents for the cleavage of disulfide bonds [61]. These mucolytic agents are generally encapsulated in NCs to prevent the disruption of the mucus layer over a wide area of the GI tract.

The immobilization of mucolytic enzymes on the surface of NCs is another strategy for designing mucolytic systems, and the application of papain, bromelain, and trypsin to the mucolytic agent for mucopenetrating NCs has been investigated [41,62,63]. This approach seemed useful for localizing the cleavage of the mucosal layer because the cleavage area is limited to the site where the NCs diffuse. In a previous study, papain-conjugated poly(acrylic acid) (PAA) NCs exhibited 2.5-fold higher mucopenetrating potential than control NCs without surface conjugation [11]. Samaridou et al. compared the mucopenetration efficiency of trypsin-, papain-, and bromelain-conjugated PLGA NCs in porcine mucus and found 2-, 3-, and 3-fold increases in permeability, respectively, compared with control NCs [41]. Although this strategy can improve the mucopenetration of NCs according to the previous reports, these enzymes can be easily deactivated in harsh environments in the GI tract (e.g., highly acidic conditions in the stomach) and proteases can degrade the enzymes. This indicates the necessity for stabilization and protection from these factors using other DDS strategies.

### 3.2. Mucoadhesive Nanoparticles

Mucoadhesive systems are thought to prolong the residence time of NCs within the mucosal layer through interactions with the mucosal components. Because of the improved residence time near the absorption site, NCs with mucoadhesive potential can provide enhanced and/or sustained drug absorption. As described in Section 2.2, mucoadhesive properties can be achieved by chain entanglements with mucin molecules and by either non-covalent interactions, such as hydrogen bonding and ionic interactions, or covalent bonds, such as disulfide bonds, between mucoadhesive polymers with thiol groups and cysteine-rich subdomains of mucus glycoproteins [64]. In this section, some key characteristics of the mucoadhesive polymers are discussed for the suitable selection of carrier materials (Table 2).

#### 3.2.1. Cationic and Anionic Charged Surface

To prepare highly mucoadhesive NCs, either exclusively cationic or anionic polymers should be selected as the surface materials to strengthen the electrostatic interactions between the mucoadhesive polymer and the mucin molecule. Chitosan, alginate, PAA, and cellulose derivatives have been widely investigated as cationic and anionic polymers for the development of NCs with mucoadhesive potential.

The cationic surface can interact with the negatively charged site of mucin derived from sialic groups by electrostatic interactions, strengthening the bond between the NCs and the mucus layer and providing greater resistance against dislodging forces [65]. Thus, enhanced bioadhesive potential can prolong the gastric residence time for long-lasting oral absorption. Chitosan is a semisynthetic polymer produced by the deacetylation of chitin and is applicable to various types of DDS carriers, such as mucoadhesive NCs, owing to its unique characteristics and high biocompatibility [66,67,68,69]. In a previous report, the promotion of membrane permeability was attributed to interactions with the membrane surface, resulting in the opening of epithelial tight junctions [70]. There are some reports on chitosan-based NCs for the oral delivery of not only small molecules but also macromolecules such as peptides like insulin; this may be possible due to their mucoadhesive and membrane-permeable potentials [71,72,73]. The combination with other biocompatible polymers, such as PEG, could also improve the mucoadhesiveness of chitosan-based NCs [74,75].

The anionic surface derived from the carboxyl groups in the monomer has the potential to form hydrogen bonds, hydrophobic interactions, and van der Waals bonds with mucosal components such as sialic acid groups and sulfate residues within the oligosaccharide chains of mucin proteins, which are controlled by the pH and ionic composition [16]. Although there are many reports on anionic mucoadhesive polymers, alginate and PAA have been extensively investigated for their potential in mucoadhesive systems owing to their high biocompatibility [76,77]. Synthetic derivatives of PAA are negatively charged polymers with mucoadhesive properties owing to their carboxyl group [78]. PAA-based materials were first synthesized and patented in 1957; several forms are available with different molecular weights and polymer architectures for use as the carrier material for oral DDSs [79]. Owing to their pH-sensitive ionization characteristics, these polymers are attractive for the localized delivery of NCs depending on the environmental pH and their long-term absorption from absorption sites [79,80]. Alginate is a polysaccharide extracted from seaweed that consists of 1–4 linked *α*-L-guluronic acid and *β*-D-mannuronic acid residues [81]. Similar to PAA, the carboxyl groups within the alginate structure can form hydrogen bonds with the sialic acid and sulfate residues in mucin, contributing to relatively strong mucoadhesive properties [82,83,84]. There are some advantages to using alginate as a carrier material; for example, it shows stronger mucoadhesion than non-ionic polymers and polycationic polymers and has biodegradable properties [85]. Hyaluronic acid and chondroitin sulfate have also been used to design mucoadhesive NCs with anionic surface charges [86,87,88].

#### 3.2.2. Formation of Disulfide Bonds

The formation of covalent bonds between the surface of the NCs and mucin molecules can theoretically contribute to achieving stronger adhesive properties than noncovalent bonds. Therefore, a lot of attention has been paid to mucoadhesive NCs with the disulfide bond-forming potential owing to the existence of a thiol group-rich domain in mucin structures [89]. For this purpose, various thiolated polymers have been developed, including thiolated-chitosan [72,90,91,92], -PAA [93,94], and -alginate [95,96]. Four types of thiolated chitosan have been reported: chitosan-cysteine, chitosan-thioglycolic acid, chitosan-thioethylamidine, and chitosan-4-thiobutyl-amidine [97]. Thiolated chitosan can strengthen the molecular interactions with mucin by forming two strong interactions: the electrostatic interactions between cationic amino moieties of chitosan and the negatively charged sialic acid of mucin, and the disulfide bond formed with cysteine-rich moieties in the mucin proteins. In a previous report, surface-modified NCs with thiolated chitosan exhibited two-fold higher mucoadhesive properties than NCs covered with non-thiolated chitosan [98]. Other thiolated mucoadhesive polymers with anionic and non-ionic properties have also been reported to exhibit improved adhesion [99,100].

For the development of mucoadhesive NCs with thiolated surfaces, appropriate reactivity should be considered [101]; high thiol reactivity is not necessary for better delivery of NCs. Generally, excessively high reactivity could cause the extensively quick formation of disulfide bonds only with the surface of the mucus layer, suggesting a poor interpenetration process during the mucoadhesion of NCs. 

**Table 2 pharmaceutics-15-02708-t002:** List of carrier materials to develop mucoadhesive NCs for oral delivery.

Carrier Materials	Mechanism of Mucoadhesion	Types of NCs	Target Drug: Outcomes	Ref.
Chitosan	Ionic interactions/hydrogen bond	PLGA nanoparticles	Diosmin: high storage stability, sustained release, ↑ gastric retention, ↑ anti-ulcer activity	[67]
Chitosan/Lecithin	Ionic interactions/hydrogen bond	Chitosan-lecithin nanocomplex	Raloxifene: low cytotoxicity, sustained release, opening tight junctions, ↑ oral absorption	[68]
Thiolated chitosan	Ionic interactions/hydrogen bond/disulfide bond	HPMCP nanoparticle	Low-molecular weight heparin: pH-responsive sustained release, ↑ pharmacodynamic action	[92]
Thiolated chitosan	Ionic interactions/hydrogen bond/disulfide bond	Liposome	Calcitonin: ↑ cellular uptake, ↑ pharmacodynamic actions	[102]
Chitosan/Chitosan-glutathione	Ionic interactions/hydrogen bond/disulfide bond	PBCA nanoparticles	Tymopentin: sustained release, stabilization of inner compound, ↑ intestinal retention in ex vivo and in vivo experiments	[69]
PAA	Hydrogen bond	Liposome	Calcitonin: ↑ pharmacodynamic action	[80]
PAA-Cys	Hydrogen bond/disulfide bond	Chitosan nanoparticles	Insulin: ↑ membrane permeation, ↑ cellular uptake, ↑ oral BA	[72]
Alginate	Hydrogen bond	PBCA nanoparticles	Insulin: sustained release, ↑ membrane permeation, ↑ oral BA	[82]
Alginate	Hydrogen bond	Chitosan nanoparticles	OVA: protection of inner drug from gastric fluid, sustained release	[83]
CSAD-VB12	Hydrogen bond	CSAD-VB12 nanoparticles	Insulin: low cytotoxicity, ↑ membrane permeation, ↑ oral absorption	[84]
S-protected thiolated fatty acid conjugate	Hydrogen bond/disulfide bond	Nanostructured lipid carrier	Bergapten: low cytotoxicity, sustained release	[100]

↑, increase/improvement; BA, bioavailability; CSAD-VB12, vitamin B12-modified amphiphilic sodium alginate derivative; HPMCP, hydroxypropyl methylcellulose phthalate; PAA, poly(acrylic acid); PAA-Cys, cysteine-conjugated PAA; PBCA, poly (*n*-butyl) cyanoacrylate; and PLGA, poly (lactic acid-*co*-glycolic acid).

The layer is rapidly cleared as part of the mucus turnover process. Thus, ideally, mucoadhesive properties should be observed at deeper sites close to the epithelial membrane after the penetration of the mucus layer. At this point, the thiolated NCs become more reactive in the deeper area of the mucus layer because the pH conditions close to the absorption membrane (pH 7.2) are more suitable for the formation of disulfide bonds by thiol-disulfide exchanging reactions than the pH conditions on the surface of the mucus layer.

## 4. Particle-Engineering Strategies of NCs for mDDS

Various formulation strategies exist for designing NCs tailored for mDDS. As discussed in previous sections, the modification of surface properties plays a pivotal role in governing the diffusion behavior within the target mucus layer. Furthermore, to ensure the stability of the encapsulated drug and regulate its release kinetics from NCs for pharmacokinetic control, the selection of appropriate formulation strategies should be based on the intended function of the NCs and the physicochemical properties of the drugs.

### 4.1. Polymeric Nanoparticles

Polymeric nanoparticles can be defined as colloidal particles ranging from 1 to 1000 nm, within which active pharmaceutical ingredients are encapsulated or adsorbed onto macromolecular substances, such as polymers [103]. Numerous studies have explored the strategic applications of polymeric NC systems in DDS. The targeted drug can be encapsulated within a polymeric nanomatrix through the spontaneous self-assembly of polymer materials, offering several advantages. These advantages include controlled drug release, contingent on characteristics of the polymers, stability of encapsulated compounds under in vivo conditions, resilience during storage, and the ability to encapsulate diverse drug modalities such as small molecular drugs, peptides, proteins, and nucleic acids [104]. The preparation of this system can be accomplished through various classical techniques, including emulsion-diffusion methods, nanoprecipitation, emulsion-coacervation, and nanoprecipitation. The surface properties of polymeric nanoparticles can be tailored by using a variety of functional block copolymers with amphiphilic properties or by chemically bonding functional polymers to the nanoparticle surface. Noteworthy examples include polymeric nanoparticles incorporating mucoadhesive and mucopenetrating polymers, such as PEG-PLGA nanoparticles [105], PEG-polystyrene (PS) nanoparticles [33], PAA-PS nanoparticles [33], hyaluronic acid-coated chitosan nanoparticles [106], hyaluronic acid-coated Eudragit S100 nanoparticles [107], zein-casein nanoparticles [108], lecithin-chitosan nanoparticles [109], chitosan-coated alginate nanoparticles [110], and poloxamer-based nanoparticles [111].

PEG-grafted poly(methacrylic acid) with wheat germ agglutinin was recently applied as a novel mucoadhesive material, resulting in a 2-fold increase in adhesive properties compared with non-functionalized samples [112]. Additionally, zwitterionic poloxamer analog-coated PLGA nanoparticles were designed to achieve effective oral delivery of insulin and exhibited enhanced oral absorption compared to poloxamer-coated PLGA nanoparticles [111]. Recently, various types of functional polymers have been developed to achieve desirable oral DDS, but careful consideration of the safety aspects is necessary for further development. Most studies mainly focus only on the actual pharmacological actions of developed systems. However, details and possible mechanisms of efficient delivery, pharmacokinetic behaviors of NCs, and biodistributions should also be carefully investigated to estimate their toxic potential.

### 4.2. Lipid-Based Nanoparticles

Recently, lipid-based nanoparticles such as liposomes, solid lipid nanoparticles (SLNs), and nanostructured lipid carriers (NLCs) have attracted increasing attention due to their potential as drug carriers. Since NCs are composed of physiological lipids, these systems offer several advantages, including high biocompatibility, controlled release based on the nature of natural lipids, and less susceptibility to erosion phenomena compared to polymeric NCs.

Liposomes can be defined as spherical vesicles consisting of an inner aqueous sinus surrounded by one or multiple homocentric lipid bilayers [113]. In liposomal systems, both hydrophilic and hydrophobic compounds can be encapsulated within the inner water phase and the lipid layer, respectively, enabling diverse drug applications. On a laboratory scale, liposomes can be prepared using various methods, such as film hydration, reversed-phase evaporation, detergent dialysis, and microfluidic techniques [114]. However, continuous mass production and quality control using these methods still pose limitations. Additionally, stability concerns in the GI tract, due to the wide range of pH conditions from the stomach to the intestine and enzymatic activity, present significant obstacles to oral delivery via liposomal systems. SLNs are lipid-based NCs that remain in a solid state at ambient and body temperatures. Physiological lipids, including glyceride mixtures, fatty acids, and steroids, can be carrier materials, and they are stabilized by biocompatible surfactants. SLNs are promising NCs to protect labile drugs as well as control/sustain the release of incorporated molecules due to their low toxicity and superior physical stability compared to other lipid-based systems. This results in improved physicochemical and biopharmaceutical properties. In spite of the advantages, the low loading efficiency of hydrophilic drugs and the possible expulsion of drugs during storage are still problems to be considered [115]. Additionally, NLCs have unique physicochemical properties and are formulated using a combination of solid and liquid lipids, thereby leading to less ordered structures with the firm inclusion of target molecules. Due to the flexible structure derived from liquid lipids, NLCs can achieve higher loading capacity along with long shelf storage than other conventional lipid-based systems. However, there are some unresolved problems related to quality challenges, like physical stability against heat stress and polymorphic changes in the lipids [116].

To introduce additional functionalities like enhanced mucopenetration, mucoadhesion, and membrane permeation, researchers often explore surface property modifications and optimize the compositions, thereby frequently incorporating functional excipients. A main strategy for achieving mucodiffusive liposomes is the use of PEG-coated lipid-lipid-based carriers. For example, Tahara et al. reported the development of PEGylated liposomes (PEG2000) with mucopenetrating ability, which demonstrated superior mucopenetration compared to liposomes modified with glycol chitosan in in vitro experiments [102]. PEG-lipid-based SLN showed improved oral bioavailability of curcumin with good mucus permeability [117]. Similarly, a chitosan-thioglycolic acid-coated liposomal formulation enhanced the intestinal absorption of insulin by promoting mucus layer and cellular membrane permeation [118]. Lipid-based NCs with surface modifications can enhance the oral bioavailability of encapsulated drugs through several mechanisms, including protection from presystemic metabolism and degradation in the GI tract and increased contact and diffusion across the mucosal and epithelial layers [119].

### 4.3. Emulsions and SEDDS

Emulsions, especially oil-in-water emulsions, and SEDDS have traditionally served as solubilization technologies for enhancing the oral bioavailability of poorly water-soluble compounds [120]. Oil-in-water emulsions consist of small oil droplets dispersed in an aqueous medium, with each droplet being coated by a thin layer of emulsifier molecules. SEDDS consist of isotropic mixtures of oils, surfactants, and cosurfactants, which spontaneously form emulsions when they come into contact with an aqueous medium. This formulation system is easily prepared by mixing all components, making it highly manufacturable for large-scale production without the need for complex particle size reduction techniques [121]. The emulsification process occurs spontaneously due to the presence of surfactants and/or cosurfactants, which reduce the interfacial tension between the oil and water phases [122].

These systems can be efficiently used to deliver lipophilic drugs categorized into biopharmaceutics classification system classes 2 and 4 owing to their solubilization potential by dissolving drugs in the oil phase and preventing precipitation in the GI tract [123]. Beyond solubilization, emulsions and SEDDS can also inhibit efflux transporters within the epithelial membrane, such as P-glycoprotein, through the use of PEG-based surfactants, including *d*-tocopheryl poly(ethylene glycol) (TPGS) [124], polysorbate 80 [125], polyoxyethylene 40 stearate [126], and cremophor EL [127]. Emulsions with PEG-coated surfaces can protect against enzymatic degradation and improve the dispersion and diffusion properties within the mucous layer. Unlike solid nanoparticle systems, emulsions enhance mucus permeation due to the flexible nature of fluidic droplets and the highly hydrophilic character of their surfaces [128]. To control the mucoadhesive property of emulsions and SEDDS, thiolated polymer and preactivated thiomer were applied in a previous study [129,130] because of the mucoadhesive properties of thiol groups. In this study, the thiomer-SEDDS exhibited lower mucopenetration and higher retention in mucus as compared to that in uncoated SEDDS, possibly due to the formation of disulfide bonds. To further improve mucopenetration properties, mucolytic SEDDS have been developed [62,131,132]. Although emulsion-based systems can be a promising strategy for improving the biopharmaceutical properties of various drugs, it is crucial to carefully consider the choice of excipients, as it considerably impacts the properties of formulations, including release rates and colloidal/storage stability. Additionally, safety concerns related to surfactants, such as potential irritation of GI membranes should be considered.

## 5. Safety Concerns of Oral NCs

As described in the above sections, mucopenetrating and mucoadhesive NCs have various advantages for oral DDS; however, nanoparticles have a wide range of safety concerns depending on their physicochemical properties including size, shape, surface properties, characteristics of materials, and biodegradability of NCs [133]. The size and shape of nanoparticles significantly influence toxicity due to the varying diffusion properties within mucus and the frequency of cellular uptake by endocytosis [134,135]. Especially, mucopenetrating NCs could have higher risks of nanotoxicity than mucoadhesive NCs due to their potential to avoid biological barriers based on the mucus layer, interact with the cellular surface of the intestinal epithelium, and enter into the systemic circulation by a translocation process across the cellular membrane. The entrance of NCs into systemic circulation might induce unexpected nanotoxicity by accumulation in various tissues and interaction with proteins and cellular membranes [136]. In previous studies, various kinds of polymers, lipids, and macromolecules have been used as carrier materials to design functional NCs. Although most of these compounds are used as excipients for other pharmaceutical applications or “Generally Recognized as Safe (GRAS)” chemicals, formulization as nanoparticles could induce toxicity depending on the physicochemical properties of carrier materials. For example, chitosan and its derivatives are widely known as safe. However, there are some reports on the toxic potential of chitosan-based nanoparticles, since the cationic nature is considered more toxic than the anionic nature owing to the high capability of interactions with negatively charged cellular membranes [137]. The biodegradability of carrier materials is an important factor in estimating polymer accumulation. Moreover, it is essential to consider the clearance of degradants [133].

Although there have been several reports on the toxicity of inorganic nanomaterials, oral NCs, such as polymeric nanoparticles, lipid-based NCs, and other NCs have received less attention. The “organic” materials used in these NCs like polymers, lipids, proteins, and polysaccharides have been considered relatively safe materials since they are employed in healthcare and pharmaceutical products. However, the nanosization of these materials would increase the safety concern. Nano-sized particles have a large surface area, which increases the chances of direct contact with the body tissues, and the extremely small particles could show unexpected pharmacokinetic behavior, possibly due to crossing the physiological barriers. Considering these points, pharmacokinetic behavior, biodistribution, and accumulation of NCs should be carefully investigated to develop effective and safe products. However, there is a lack of basic information on the biological behaviors of NCs under in vivo conditions. Attention must be paid to these issues to maximize the potential of functional NCs.

## 6. Future Perspectives and Conclusions

Oral administration is the preferred route for drug delivery due to its numerous advantages, including good patient compliance and effective medication delivery. Therefore, there have been numerous efforts to develop various types of oral DDS. However, there are significant challenges in achieving efficient drug absorption from the GI tract, including the pH gradient from the stomach to the colon, metabolic enzymes, the presence of a mucus layer on the surface of epithelial cells, and permeation of the epithelial cellular membrane. In addressing these issues, nanoparticle systems have been extensively studied.

Although conventional NCs have been explored for oral DDS, primarily focusing on improving drug dissolution and controlled release, they may not adequately overcome the physiological barriers that affect pharmacokinetic control after oral administration. To address these challenges, surface modification of NCs can enhance their effectiveness in delivering drugs orally by incorporating mucopenetrating and mucoadhesive functionalities. As a result, there is a growing interest in the development of novel technologies for appropriate surface modification. Mucopenetrating NCs have the unique ability to swiftly transport encapsulated drugs to the epithelial membrane. This capability is attributed to their bioinert surface, which is coated with neutrally charged polymers and mucolytic agents. A highly bioinert surface is essential for achieving superior penetration properties. However, it can also be inert to the epithelial membrane, potentially leading to reduced cellular uptake. In contrast, mucopenetrating NCs can extend their presence within the body by forming interactions with mucin, enhancing oral absorption over an extended period. Nevertheless, there remains the risk of rapid clearance from the mucus layer’s surface due to the continuous turnover of the mucus layer. Therefore, developing NCs with environmentally responsive surfaces holds promise for the advancement of mDDS to achieve more effective delivery of target compounds [138]. These systems can switch their surface charge from negative to positive within the mucus gel layer, thereby closely interacting with the underlying epithelium. This dynamic interaction provides both mucopenetrating properties and strong bonds with mucin and the epithelial membrane in the deeper regions of the mucus layer. Although novel polymers have been synthesized to design charge-converting NCs in some studies, it is imperative to gain a deeper understanding of their potential toxicity and optimize these systems for future clinical trials. Additionally, developing suitable evaluation systems for assessing physicochemical properties and biodistribution within mucus layers is essential for designing effective oral NCs. Recent advancements in imaging technologies have enabled researchers to visualize the dispersion behaviors and distributions of nanoparticles within the GI tract. This progress has contributed to a deeper understanding of the fate of NCs within the GI tract. Utilizing techniques like fluorescence resonance energy transfer (FRET) systems and environmentally responsive fluorescent probes, such as aggregation-causing quenching (ACQ) probes, offers a promising approach to visualization [139].

This review primarily focuses on controlled drug absorption from the GI tract. However, mDDS can be applied to various administration routes with mucous layers, including the eyes, nose, mouth, airway, and lungs. Depending on the physiological conditions and the disease at the target site with mucous layers, the characteristics of the mucous layers differ significantly in terms of volume, viscosity, turnover, and the pore size of the mucin mesh. This diversity underscores the need for an appropriate design of NCs. Recently, there has been an increasing number of reports on the development of mucopenetrating and mucoadhesive NCs for systemic delivery and topical treatment of diseases at these administration sites. Additionally, diverse nanocarrier-based DDS, such as polymeric nanoparticles, liposomes, solid lipid nanoparticles, and SEDDS, enable the encapsulation of various types of drugs, including small molecules, peptides, nucleic acids, and proteins. Consequently, functional NCs can significantly contribute to the efficient and effective treatment of numerous diseases, and appropriate final dosage forms should be selected for the optimal treatment of target diseases. Nonetheless, safety concerns persist due to the materials used and the unclear pharmacokinetic behavior of these novel NCs. Although biodegradable materials may pose limited risks, careful consideration of excipient selection and long-term safety evaluations are necessary. To develop ideal NCs for specific target diseases, close communication and collaboration with formulators, physiologists, and toxicologists are imperative, all while keeping in mind the relevant safety regulations.

In conclusion, several recent studies on NCs with controlled diffusion properties in the mucus layer have suggested that surface-modified NCs can significantly improve the mucopenetration and mucoadhesive properties of various drug modalities. However, the risk of side effects and the difficulty of scaling up most surface modification techniques remain major obstacles to their application to commercial products. Additionally, regulatory issues of pharmaceutical excipients are considered to accelerate the clinical application of novel technologies. Although there are many reports on functional NCs, the approved and commercially available NC systems are quite limited. If these hurdles can be overcome through clinical studies and optimization of the manufacturing process, mDDS-based NCs may open a bright future for the treatment of various diseases.

## Figures and Tables

**Figure 1 pharmaceutics-15-02708-f001:**
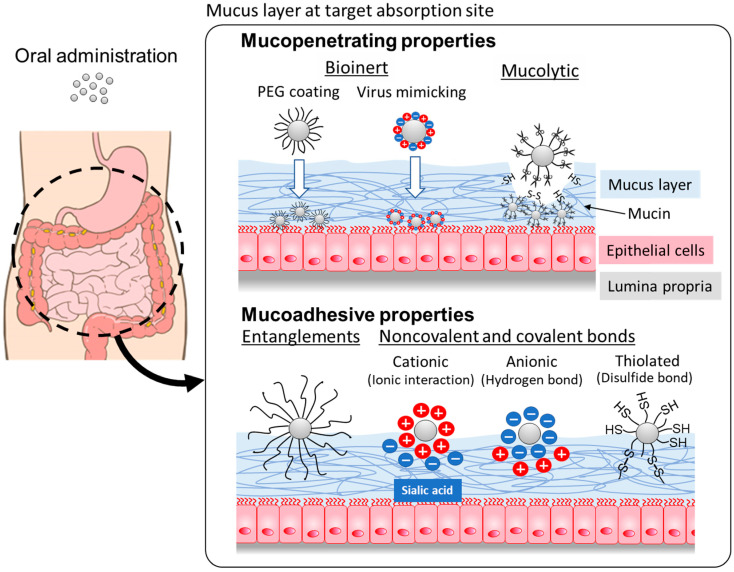
Schematic illustrations of the mechanisms for mucopenetrating and mucoadhesive properties of NCs after oral administration.

**Table 1 pharmaceutics-15-02708-t001:** List of carrier materials to develop mucopenetrating NCs for oral delivery.

Polymers	Mechanism of Mucopenetration	Types of NCs	Target Drug: Outcomes	Ref.
DSPE-PEG 2000	PEG surface	PLGA nanoparticles/ Lipoid S100	Silibinin: ↑ cell internalization, ↑ oral BA	[30]
PLA-PEG	PEG surface	Mesoporous silica nanoparticles	Insulin: high loading, ↑ cellular uptake by caveolae-mediated endocytosis, ↑ pharmacodynamic action	[31]
Pluronic F127	PEG surface	Liposome	Cyclosporine A: stabilization of liposome in simulated GI conditions, ↑ oral BA	[32]
PS-PEG	PEG surface	PS-PEG nanoparticles	Cyclosporine A: ↑ dissolution of cyclosporine A, ↑ oral BA	[33]
Vit E-PEG 5000	PEG surface	PLGA nanoparticles	Paclitaxel: high loading, sustained release	[34]
DLPC	Zwitterionic surface	Mesoporous silica	Insulin: ↑ cellular uptake, ↑ oral absorption	[35]
DLPC	Zwitterionic surface	PLA nanoparticles	Insulin: ↑ affinity to cellular membrane, ↑ oral absorption	[36]
			Low toxicity in in vitro and in vivo evaluations	[37]
Betaine polymer	Zwitterionic surface	Micelle/nanogel	Insulin: improved cellular uptake without opening tight junction, ↑ oral BA	[38]
Dodecyl sulfobetaine	Zwitterionic surface	Porous silicon nanoparticles	Insulin: ↑ cellular membrane permeability, ↑ pharmacodynamic action	[39]
NAPG	Mucolytic	Nanostructured lipid carrier	Curcumin: high encapsulation efficiency, ↑ oral BA	[40]
Papain/bromelain-conjugated PAA	Mucolytic	PAA nanoparticles	Increase the mobility of mucus, breaking the mucin structure	[41]

↑, increase/improvement; BA, bioavailability; DLPC, 1,2-Dilauroyl-sn-glycero-3-phosphorylcholine; DSPE-PEG, 1,2-distearoyl-sn-glycero-3-phosphoethanolamine-*N*-[amino(polyethylene glycol)-2000]; NAPG,N-acetyl-L-cysteine-polyethylene glycol (100)-monostearate; PAA, poly(acrylic acid); PLA, poly(lactic acid); PLA-PEG, poly D,L,-lactic acid-polyethylene glycol block copolymer; PLGA, poly (lactic acid-*co*-glycolic acid); PS-PEG, polystyrene-poly(ethylene glycol) graft copolymer; and VitE-PEG5000, vitamin E conjugated PEG 5000.

## Data Availability

Not applicable.

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
