# Peer review of "Recent Advancements in the Development of Nanocarriers for Mucosal Drug Delivery Systems to Control Oral Absorption"

_pharmaceutics, 2023, doi:10.3390/pharmaceutics15122708_

Round 1

Reviewer 1 Report

Comments and Suggestions for Authors

In this review, the physicochemical properties and physiological barriers of gastrointestinal mucosa have been comprehensively summarized. According to the properties of GI tract, the required functions of nanocarriers are summarized in detail. There are two main characteristics of the nanocarriers, including mucous adhesion and mucous permeability, and several commonly used nanocarriers are discussed. This review provides an in-depth discussion on the functional mechanisms of different nanocarriers to achieve mucous permeability and mucous adhesion respectively, contributing to design more perfect functional nanocarriers for oral drug delivery in the future.

However, there are still some concerns in this review that can be improved.

1. Liposomes, solid lipid nanoparticles, and nanostructured lipid carriers are all lipid-based drug delivery systems. It is not comprehensive that the authors only list liposome as a kind of particle engineering strategies, please group these formulations together for introduction.

2. Similarly, microemulsions or nanoemulsions should be introduced along with SEDDS.

3. It is suggested to increase figures and tables to improve the readability of the review.

Comments on the Quality of English Language

The overall expression of the review is relatively smooth and clear, but there are still a little problems to concern:

1. There are many repetitive expressions. So the logic of this review needs to be strengthened to achieve simple and clear expression.

2. There are some language mistakes, non-native expression errors and garbled codes of units, please pay attention to them.

Author Response

Comment 1 “In this review, the physicochemical properties and physiological barriers of gastrointestinal mucosa have been comprehensively summarized. According to the properties of GI tract, the required functions of nanocarriers are summarized in detail. There are two main characteristics of the nanocarriers, including mucous adhesion and mucous permeability, and several commonly used nanocarriers are discussed. This review provides an in-depth discussion on the functional mechanisms of different nanocarriers to achieve mucous permeability and mucous adhesion respectively, contributing to design more perfect functional nanocarriers for oral drug delivery in the future. However, there are still some concerns in this review that can be improved.”

We are really grateful for the expert comments and excellent advice we have received.  The thoughtful comments have helped us to strengthen and improve the discussion of the manuscript.  We have revised the manuscript thoroughly as the reviewer kindly pointed out.  We hope our responses are still satisfactory ones for you.

Comment 2 “ Liposomes, solid lipid nanoparticles, and nanostructured lipid carriers are all lipid-based drug delivery systems. It is not comprehensive that the authors only list liposome as a kind of particle engineering strategies, please group these formulations together for introduction.”

              We completely agree with the comment.  As the reviewer kindly pointed out, using “lipid-based nanoparticles” as the section title and summarizing the lipid-based DDS are appropriate to make the section comprehensive.  The manuscript has been revised according to the reviewer’s suggestion as follows; “4.2. Lipid-based nanoparticles

Recently, lipid-based nanoparticles such as liposomes, solid lipid nanoparticles (SLNs), and nanostructured lipid carriers (NLCs) have attracted increasing attention due to their potential as drug carriers. Since NCs are composed of physiological lipids, these systems offer several advantages, including high biocompatibility, controlled release based on the nature of natural lipids, and less susceptibility to erosion phenomena com-pared to polymeric NCs.

Liposomes can be defined as spherical vesicles consisting of an inner aqueous sinus surrounded by one or multiple homocentric lipid bilayers [112].” (Page 11, Line 454); “SLNs are lipid-based NCs that remain in a solid state at ambient and body temperatures. Physiological lipids, including glyceride mixtures, fatty acids, and steroids, can be the carrier materials, and they are stabilized by biocompatible surfactants. SLNs are promising NCs to protect labile drugs as well as control/sustain the release of incorporated molecules due to their low toxicity and superior physical stability compared to other lipid-based systems. This results in improved physicochemical and biopharmaceutical properties. In spite of the advantages, the low loading efficiency of hydrophilic drugs and the possible expulsion of drugs during storage are still problems to be considered [114]. Additionally, NLCs have unique physicochemical properties and are formulated using a combination of solid and liquid lipids, thereby leading to less ordered structures with the firm inclusion of target molecules. Due to the flexible structure derived from liquid lipids, NLCs can achieve higher loading capacity along with long shelf storage than other conventional lipid-based systems. However, there are some unresolved problems related to quality challenges, like physical stability against heat stress and polymorphic changes in the lipids [115].

To introduce additional functionalities like enhanced mucopenetration, mucoadhesion, and membrane permeation, researchers often explore surface property modifications and optimize the compositions, thereby frequently incorporating functional excipients. A main strategy for achieving mucodiffusive liposomes is the use of PEG-coated lipid-lipid based carriers.” (Page 12, Line 470); “PEG-lipid based SLN showed improved oral bioavailability of curcumin with good mucus permeability [117].” (Page 12, Line 489); and “Lipid-based NCs with surface modifications can enhance the oral bioavailability of encapsulated drugs through several mechanisms, including protection from presystemic metabolism and degradation in the GI tract and increased contact and diffusion across the mucosal and epithelial layers [119].” (Page 12, Line 495)

Comment 3 “Similarly, microemulsions or nanoemulsions should be introduced along with SEDDS.”

              We thank for the comment.  As the reviewer’s suggestion, emulsion systems has been also introduced in the section as follows; “4.3. Emulsions and SEDDS

Emulsions, especially oil-in-water emulsions, and SEDDS have traditionally served as solubilization technologies for enhancing the oral bioavailability of poorly water-soluble compounds [120]. Oil-in-water emulsions consist of small oil droplets dispersed in an aqueous medium, with each droplet being coated by a thin layer of emulsifier molecules.” (Page 12, Line 500); “These systems can be efficiently used to deliver lipophilic drugs categorized into biopharmaceutics classification system classes 2 and 4 owing to their solubilization potential by dissolving drugs in the oil phase and preventing precipitation in the GI tract [123].” (Page 12, Line 512); and “To control the mucoadhesive property of emulsions and SEDDS, thiolated polymer and preactivated thiomer were applied in a previous study [129, 130] because of the mucoadhesive properties of thiol groups.” (Page 13, Line 522)

Comment 4 “It is suggested to increase figures and tables to improve the readability of the review.”

              We appreciate the invaluable comment from the reviewer.  Although we considered to increase the number of figures and tables in the manuscript, we think that minimum essential figure and tables to understand the main point are already shown in the manuscript.

Comment 5 “The overall expression of the review is relatively smooth and clear, but there are still a little problems to concern:

  1. There are many repetitive expressions. So the logic of this review needs to be strengthened to achieve simple and clear expression.
  2. There are some language mistakes, non-native expression errors and garbled codes of units, please pay attention to them.”

              We appreciate the invaluable comments from the review.  To solve the problems pointed out by the reviewer, the manuscript has been checked thoroughly and revised throughout the manuscript and checked again by a native speaker with scientific background.

Reviewer 2 Report

Comments and Suggestions for Authors

The current work focuses on Recent advancements in developing nanocarriers for mucosal drug delivery systems to control oral absorption. The author’s great effort into the manuscript, but minor issues should be addressed. 

-Toxicity is significant for medical application subsection should be inserted related to the toxicity of formulation for designing NCs tailored for mDDS.

-There is a felt lack of critical assessments by the authors. The authors did not mention the research gap between the previously reported articles and the present situation. Authors should incorporate their views in each subsection to mold the research in a new direction.

Conclusions and future remarks

Specific points for Future Perspectives and Conclusive Remarks are required

References

The reference title is missing

Author Response

Comment 1 “The current work focuses on Recent advancements in developing nanocarriers for mucosal drug delivery systems to control oral absorption. The author’s great effort into the manuscript, but minor issues should be addressed.”

We are really grateful to the reviewer for invaluable comments.  The reviewer's suggestion would be helpful for us to strengthen the discussion of our manuscript.  We have revised the manuscript thoroughly as the reviewer pointed out.  We sincerely hope that our responses will be satisfactory one’s for you.

Comment 2 “Toxicity is significant for medical application subsection should be inserted related to the toxicity of formulation for designing NCs tailored for mDDS.”

              We completely agree with the comment.  As the reviewer suggested, the section of “Safety concerns of oral NCs” has been added in the manuscript as follows; “5. Safety concerns of oral NCs

As described in the above sections, mucopenetrating and mucoadhesive NCs have various advantages for oral DDS; however, nanoparticles have a wide range of safety concerns depending on their physicochemical properties including size, shape, surface properties, characteristics of materials, and biodegradability of NCs [133]. The size and shape of nanoparticles significantly influence toxicity due to the varying diffusion properties within mucus and the frequency of cellular uptake by endocytosis [134, 135]. Especially, mucopenetrating NCs could have higher risks of nanotoxicity than mucoadhesive NCs due to their potential to avoid biological barriers based on the mucus layer, interact with the cellular surface of the intestinal epithelium, and enter into the systemic circulation by a translocation process across the cellular membrane. The entrance of NCs into systemic cirtulation might induce unexpected nanotoxicity by accumulation in various tissues and interaction with proteins and cellular membranes [136]. In previous studies, various kinds of polymers, lipids, and macromolecules have been used as carrier materials to design functional NCs. Although most of these compounds are used as excipients for other pharmaceutical applications or “Generally Recognized as Safe (GRAS)” chemicals, formulization as nanoparticles could induce toxicity depending on the physicochemical properties of carrier materials. For example, chitosan and its derivatives are widely known as safe. However, there are some reports on the toxic potential of chitosan-based nanoparticles, since the cationic nature is considered to be more toxic than the anionic nature owing to the high capability of interactions with negatively charged cellular membranes [137]. The biodegradability of carrier materials is an important factor in estimating polymer accumulation. Moreover, it is essential to consider the clearance of degradants [133].

Although there have been several reports on the toxicity of inorganic nanomaterials, oral NCs, such as polymeric nanoparticles, lipid-based NCs, and other NCs have received less attention. The “organic” materials used in these NCs like polymers, lipids, proteins, and polysaccharides have been considered relatively safe materials since they are employed in healthcare and pharmaceutical products. However, nanosization of these materials would increase the safety concern. Nano-sized particles have a large surface area, which increases the chances of direct contact with the body tissues, and the extremely small particles could show unexpected pharmacokinetic behavior, possibly due to crossing the physiological barriers. Considering these points, pharmacokinetic behavior, bio-distribution, and accumulation of NCs should be carefully investigated to develop effective and safe products. However, there is a lack of basic information on the biological behaviors of NCs under in vivo conditions. Attention must be paid to these issues to maximize the potential of functional NCs.” (Page 13, Line 530)

Comment 3 “There is a felt lack of critical assessments by the authors. The authors did not mention the research gap between the previously reported articles and the present situation. Authors should incorporate their views in each subsection to mold the research in a new direction.”

              We thank for the comment from the reviewer.  As per the suggestion, the research gaps between the previous reports and the present situation have been described in appropriate sections as follows; “According to previous reports, the PEG-coated surface enables NCs to penetrate the mucus layer quickly; however, a highly bioinert surface also reduces the interaction with the cellular membrane at the absorption site, thereby possibly limiting the absorption process if free drug molecules cannot be released from the NCs. Additionally, the cellular uptake of PEG-coated NCs, especially those coated with long PEG chains, can be limited because of steric hindrance, reduced surface charge, and hydrophilic-hydrophobic repulsion [45]. Therefore, it is necessary to consider the balance between the bioinertness with mucin and its interactions with cellular membranes to control the absorption process using PEG-coated NCs. In recent years, there have been some reports on the appearance of PEG antibodies due to the excessive use of PEG for DDS applications [46], thereby causing the alteration of mobility and biodistribution of PEGylated nanoparticles in mucus [47]. Thus, developing an alternative bioinert polymer might be necessary for designing desirable oral NCs.” (Page 7, Line 242); “Despite its increasing number of applications on zwitterionic NCs in recent years, the potential for mucopenetration remains unexplored and possible mechanisms are still un-clear. Thus, a deeper understanding of mucopenetrating mechanisms by zwitterionic NCs could help select the appropriate components in functional NCs.” (Page 8, Line 285); “Although this strategy can improve the mucopenetration of NCs according to the previous reports, these enzymes can be easily deactivated in harsh environments in the GI tract (e.g., highly acidic conditions in the stomach) and proteases can degrade the enzymes. This indicates the necessity for stabilization and protection from these factors using other DDS strategies.” (Page 8, Line 322); “Recently, various types of functional polymers have been developed to achieve desirable oral DDS, but careful consideration of the safety aspects is necessary for further development. Most studies mainly focus only on the actual pharmacological actions of developed systems. However, details and possible mechanisms of efficient delivery, pharmacokinetic behaviors of NCs, and biodistributions should also be carefully investigated to estimate their toxic potential.” (Page 11, Line 447); and “Although there have been several reports on the toxicity of inorganic nanomaterials, oral NCs, such as polymeric nanoparticles, lipid-based NCs, and other NCs have received less attention. The “organic” materials used in these NCs like polymers, lipids, proteins, and polysaccharides have been considered relatively safe materials since they are employed in healthcare and pharmaceutical products. However, nanosization of these materials would increase the safety concern. Nano-sized particles have a large surface area, which increases the chances of direct contact with the body tissues, and the extremely small particles could show unexpected pharmacokinetic behavior, possibly due to cross-ing the physiological barriers. Considering these points, pharmacokinetic behavior, bio-distribution, and accumulation of NCs should be carefully investigated to develop effective and safe products. However, there is a lack of basic information on the biological behaviors of NCs under in vivo conditions. Attention must be paid to these issues to maximize the potential of functional NCs.” (Page 13, Line 557)

Comment 4 “Conclusions and future remarks Specific points for Future Perspectives and Conclusive Remarks are required”

              We thank for the comment.  The manuscript has been revised to show more specific points in Future perspectives and conclusion as follows; “Therefore, developing NCs with environmentally responsive surfaces holds promise for the advancement of mDDS to achieve more effective delivery of target compounds [138]. These systems can switch their surface charge from negative to positive within the mucus gel layer, thereby closely interacting with the underlying epithelium. This dynamic inter-action provides both mucopenetrating properties and strong bonds with mucin and the epithelial membrane in the deeper regions of the mucus layer. While novel polymers have been synthesized to design charge-converting NCs in some studies, it is imperative to gain a deeper understanding of their potential toxicity and optimize these systems for future clinical trials. Additionally, developing suitable evaluation systems for assessing physicochemical properties and biodistribution within mucus layers is essential for designing effective oral NCs. Recent advancements in imaging technologies have enabled researchers to visualize the dispersion behaviors and distributions of nanoparticles within the GI tract. This progress has contributed to a deeper understanding of the fate of NCs within the GI tract. Utilizing techniques like fluorescence resonance energy transfer (FRET) systems and environmentally responsive fluorescent probes, such as aggregation-causing quenching (ACQ) probes, offers a promising approach for visualization [139].” (Page 14, Line 588); “However, the risk of side effects and the difficulty of scaling up most surface modification techniques remain major obstacles to their application to commercial products. Addition-ally, regulatory issues of pharmaceutical excipients are considered to accelerate the clinical application of novel technologies. Although there are many reports on functional NCs, the approved and commercially available NCs systems are quite limited. If these hurdles can be overcome through clinical studies and optimization of the manufacturing process, mDDS-based NCs may open a bright future for the treatment of various diseases.” (Page 15, Line 632)

Comment 5 “References The reference title is missing”

              We apologize for missing title of reference section.  The manuscript has been revised appropriately.

Reviewer 3 Report

Comments and Suggestions for Authors

 The authors of the manuscript summarize information about the mucosal layer and then present nanocarriers suitable for drug delivery, which may be suitable for the design of mucosal drug delivery systems. The manuscript includes 2 useful tables and an informative figure. The authors have used 130 references to prepare this paper. The structure of the manuscript is logical and well structured, but some issues closely related to the topic are not addressed.

It should be considered that the authors explain how mucoadhesion can be studied up to the beginning of a few sentences. 

For nanocarriers, what are the final dosage forms that are suitable for oral delivery? This may also be worth collecting and even providing a table.

A list of abbreviations might be useful.  

Comments on the Quality of English Language

Minor editing of English language required

Author Response

Comment 1 “The authors of the manuscript summarize information about the mucosal layer and then present nanocarriers suitable for drug delivery, which may be suitable for the design of mucosal drug delivery systems. The manuscript includes 2 useful tables and an informative figure. The authors have used 130 references to prepare this paper. The structure of the manuscript is logical and well structured, but some issues closely related to the topic are not addressed.”

We really thankful to the reviewer for favorable comments, and we believe that the reviewer's suggestions would be helpful to improve the quality of our manuscript.  We have revised the manuscript accordingly as the reviewer pointed out.  We hope our responses are still satisfactory one’s for you.

Comment 2 “It should be considered that the authors explain how mucoadhesion can be studied up to the beginning of a few sentences.”

We thank for the comment.  As the reviewer’s suggestion, some sentences has been added to explain how mucoadhesion ca be studied as follows; “The mucus layer can be used to adjust the residence time of NCs by modifying the surface properties to develop mDDS-based NCs to control intestinal absorption. NCs with mucoadhesive and mucopenetrating potentials can be developed by changing the interactions between the mucin layer and surface of NCs (Figure 1). These properties could contribute to the control of the absorption process of drugs encapsulated into the NCs after oral administration. Generally, mucoadhesive NCs can extend the absorption process, which result in prolonged systemic exposure, and mucopenetrating NCs can achieve quick absorption from the absorption site.” (Page 2, Line 74)

Comment 3 “For nanocarriers, what are the final dosage forms that are suitable for oral delivery? This may also be worth collecting and even providing a table.”

              We partly agree with the comment.  Although the selection of appropriate final dosage form is important issue for clinical applications of functional NCs, the main focus of this article is the summarization and discussion of mDDS for controlling the absorption from GI tracts.  Additionally, there are very limited information of final dosage forms based on NCs systems because mDDS is relatively recent technologies for oral DDS.  Thus, we have added a sentence to show the importance of selecting appropriate final dosage form for future clinical applications in Conclusions and future perspectives; “Consequently, functional NCs can significantly contribute to the efficient and effective treatment of numerous diseases, and appropriate final dosage forms should be selected for the optimal treatment of target diseases.” (Page 14, Line 632)

Comment 4 “A list of abbreviations might be useful.”

              We completely agree with the comment.  The list of abbreviations has been added in the manuscript; “Abbreviations: AQC; aggregation-causing quenching; BA, bioavailability; CSAD-VB12, vitamin B12-modified amphiphilic sodium alginate derivative; DLPC, 1,2-Dilauroyl-sn-glycero-3-phosphorylcholine; DSPE-PEG, 1,2-distearoyl-sn-glycero-3-phosphoethanolamine-N-[amino(polyethylene glycol)-2000]; FRET, fluorescence resonance energy transfer; GI, gastrointestinal; GRAS, generally recognized as safe; HPMCP, hydroxylpropyl methylcellulose phthalate; mDDS, mucosal drug delivery systems; MUC2, mucin 2; NAC, N-acetyl cysteine; NAPG,N-acetyl-L-cysteine-polyethylene glycol (100)-monostearate; NCs, nanocarriers; NSCs, nanostructured lipid carriers; PAA, poly(acrylic acid); PAA-Cys , cysteine conjugated PAA; PBCA, poly (n-butyl) cyanoacrylate; PEG, polyethylene glycol; PLA, poly(lactic acid); PLA-PEG, poly D,L,-lactic ac-id-polyethylene glycol block copolymer; PLGA, poly (lactic acid-co-glycolic acid); PS, polystyrene; PS-PEG, polystyrene-poly(ethylene glycol) graft copolymer; SEDDS, self-emulsifying drug delivery system; SLNs, solid lipid nanoparticles; and VitE-PEG5000, vitamin E conjugated PEG 5,000” (Page 15, Line 650)

Round 2

Reviewer 1 Report

Comments and Suggestions for Authors

In the revised version, the authors have implemented significant improvements to enhance the manuscript. Notably, the inclusion of lipid-based nanoparticles and SEDDS greatly improved the integrity of the review. This review provides a detailed summary of the mechanisms underlying nanocarriers for mucosal drug delivery systems to control oral absorption. In addition, the safety of oral NC is also very necessary. The supplement of perspectives and conclusion also reflects the deeper understanding on the topic, which is very enlightening.

In summary, the logic and completeness of the present manuscript have greatly improved, making it suitable for publication consideration in Pharmaceutics.